# What Is Written on a Dog’s Face? Evaluating the Impact of Facial Phenotypes on Communication between Humans and Canines

**DOI:** 10.3390/ani13142385

**Published:** 2023-07-22

**Authors:** Courtney L. Sexton, Colleen Buckley, Jake Lieberfarb, Francys Subiaul, Erin E. Hecht, Brenda J. Bradley

**Affiliations:** 1Department of Population Health Sciences, Virginia-Maryland College of Veterinary Medicine, Virginia Polytechnic Institute and State University, Blacksburg, VA 24061, USA; 2Center for the Advanced Study of Human Paleobiology, Department of Anthropology, The George Washington University, Washington, DC 20052, USA; 3Independent Data Analyst, Washington, DC 20052, USA; 4Department of Speech, Language and Hearing Sciences, The George Washington University, Washington, DC 20052, USA; 5Department of Human Evolutionary Biology, Harvard University, Cambridge, MA 02138, USA

**Keywords:** human–canine interaction, canine communication, dogs, domestication, canine behavior, facial communication, facial markings, human–animal interaction

## Abstract

**Simple Summary:**

As dogs evolve to fill a new and increased number of roles in human societies, it is critical that we understand how they communicate with people. Here, we investigate whether markings on dogs’ faces influence how expressive they are perceived to be by humans. Using standardized systems to analyze dogs’ facial complexity and behaviors, we find that dogs with plainer faces (fewer markings) objectively score as more behaviorally expressive. Age and skill or training level also impact expressivity, with adult dogs being more expressive than senior dogs and dogs that are highly skilled being more expressive than those who have had no training or working experience. Interestingly, dogs tend to use their face more “wholistically” during highly social interactions with owners than when presented with ambiguous cues, and owners of adult dogs with plainer faces tend to be more accurate at judging their dog’s expressivity. These data are important to consider as the human–dog relationship continues to develop, both from an evolutionary perspective and especially in the context of canine training and welfare.

**Abstract:**

Facial phenotypes are significant in communication with conspecifics among social primates. Less is understood about the impact of such markers in heterospecific encounters. Through behavioral and physical phenotype analyses of domesticated dogs living in human households, this study aims to evaluate the potential impact of superficial facial markings on dogs’ production of human-directed facial expressions. That is, this study explores how facial markings, such as eyebrows, patches, and widow’s peaks, are related to expressivity toward humans. We used the Dog Facial Action Coding System (DogFACS) as an objective measure of expressivity, and we developed an original schematic for a standardized coding of facial patterns and coloration on a sample of more than 100 male and female dogs (*N* = 103), aged from 6 months to 12 years, representing eight breed groups. The present study found a statistically significant, though weak, correlation between expression rate and facial complexity, with dogs with plainer faces tending to be more expressive (r = −0.326, *p* ≤ 0.001). Interestingly, for adult dogs, human companions characterized dogs’ rates of facial expressivity with more accuracy for dogs with plainer faces. Especially relevant to interspecies communication and cooperation, within-subject analyses revealed that dogs’ muscle movements were distributed more evenly across their facial regions in a highly social test condition compared to conditions in which they received ambiguous cues from their owners. On the whole, this study provides an original evaluation of how facial features may impact communication in human–dog interactions.

## 1. Introduction

Dogs have acquired behavioral and anatomical traits that engender successful social interaction with humans. The unique interspecies relationship between humans and dogs seems an evolutionary anomaly, differing in biologically and cognitively significant ways from other instances of heterospecific mutualism and commensalism [1,2,3].

This unique relationship has influenced the bounty of the current research aiming to uncover and define the spectrum of cognitive abilities of dogs living in human societies [4], as well as how these abilities compare to those of dogs’ close taxonomic relatives (namely, wolves and other wild canids). Dogs are remarkably socially attuned to humans, attend to direct human signals, speech, and ostensive cues, and are highly trainable for a variety of tasks (e.g., herding, scent detection, medical detection, search and rescue, etc.). Investigations of the genetic bases for such abilities indicate that these traits are highly heritable between breeds and breed groups [5,6]. From a neuroanatomical perspective, historical selection by humans for working skills influences the brain structure in individual breeds [7]. Still, significant gaps in knowledge of how and why dogs arrived at their current state remain, which places us at a disadvantage in understanding the role of a critical character in our own evolutionary tale, and yet, some answers may be staring us in the face. 

For highly social humans, faces are useful for identifying individuals but are also the key to understanding, analyzing, and modifying behavior in response to the perceived thoughts, intentions, and feelings of others [8,9]. When looking at another human, our faces exhibit minute shifts in position that unconsciously mimic the feeling meant to be depicted in the other’s expression [10,11]. This automatic mimicry allows our brains to process the emotion in addition to the visual signal presented [12,13]. Additionally, according to Wood and colleagues [14], it allows us to better empathize with the emotional state of the “other”.

Indeed, across human cultures and populations, a mechanism for establishing the connections necessary for social learning and, arguably, survival comes in the form of a “universal” language grounded in visual cues that are especially reliant on the face [15,16,17], but also see Jack et al. [18].

Living closely with humans, dogs have not only evolved the ability to distinguish familiar human faces and process human facial cues, but they have also developed a propensity for responding in kind [19,20,21]. In particular, dogs make and maintain eye contact and use a variety of facial gestures to effectively communicate with human companions [21,22,23] and may even have developed facial expressions in response to non-human stimuli, such as pain [24]. They likewise understand the emotional valence of human faces [25,26]. Nagasawa and colleagues [27] show that “human-like modes of communication, including mutual gaze, in dogs may have been acquired during domestication with humans”.

While dogs may be unique in their readiness to make and keep eye contact with humans, gaze behavior is not insignificant for other canids, especially dogs’ wolf relatives. Gaze among conspecifics is typically regarded as an agonistic signal, though other facial expressions are also generally relevant for canids in some similar ways as for primates, especially those facial signals related to play [28,29]. Unlike other canids, however, wolves have facial color patterns in which gaze direction can be easily identified, which Ueda et al. [30] suggest is related to obligate group living and cooperative hunting—not unlike the adaptation of white sclerae in humans [31]. 

Receptive features of communication (e.g., markings and coloration patterns) on the faces of dogs’ close relatives (wolves) and human close relatives (other primates) may be shaped by similar forces, namely the effect they have on the intended signal receiver. Santana et al. [32] and Santana et al. [33] find that while more social primate species tend to have more facial coloration, additional research finds that among those species, individuals with plainer faces display a broader repertoire of facial expressions (“plain face phenomenon”) [34]. This pattern among fixed traits likely evolved to aid the conspecific comprehension of information contained in those more flexible, productive features of facial communication, gestures, and expressions [35]. Dogs, however, have developed highly expressive facial behaviors, including paedomorphic expressions that potentially increase the likelihood of receiving human care [36] due to social interactions with humans [21,37,38] more so than with other dogs. 

Therefore, in this study, we ask: are dogs with more complex facial features more or less behaviorally expressive toward humans than those with plainer faces? Given the heritability of temperament (including communication-related traits) across breeds [5,39], we also explored whether age, breed group, or work status impacts the dogs’ objective scores of facial expressions. Finally, we ask how accurately humans subjectively perceive canine facial expressivity compared to the objective scores. The overall aim of this study is to investigate the variations in facial markings, expressivity, and behaviors of domesticated dogs integrated into human homes and communities in order to determine whether there is a connection between superficial facial phenotypes and behavioral adaptations for communication with human social partners. 

We hypothesize that the selective pressures on the physical facial features observed in primates and wolves will be disrupted in dogs, given the history of domestication and intentional breeding in dogs. That is, due to multiple breeds with unique physical phenotypes maintaining similarly close social relationships with humans, the “plain face phenomenon” [34] observed in primates should not apply to dogs.

As an alternative hypothesis, superficial markings could serve to enhance performance and, thus, desirability for specific breed-to-task orientation. For example, on the one hand, markings on the faces of herding dogs may help to mask the visibility of facial expressions that might otherwise give away behavioral intention [40,41], which would be suitable for working sheep or cattle. On the other hand, a plain face would be more advantageous to a retriever primarily directing signals toward human hunting partners if expressions are, in fact, clearer on a plain-faced dog. However, because of the physical variation within breeds and breed groups, this alternative hypothesis would predict that the differences in behavioral rates of expression will only be observed *between* breed groups. Of course, there is also the possibility that facial markings could be unrelated to actual or perceived expressivity.

## 2. Materials and Methods

### 2.1. Ethical Considerations

The Harvard University-Area Committee approved the experiment on the Use of Human Subjects under the protocol title: Cognition, motivation, and emotion in domestic dog breeds; Harvard Principal Investigator: Erin Hecht; Protocol #: IRB19-0476/SITE20-0061. The above-named committee approved the George Washington University as a relying institution; George Washington Principal Investigator Courtney Sexton; Federal-wide Assurance: FWA00005945.

### 2.2. Subjects

*Recruitment*. Volunteers and their dogs were recruited for this project (titled “What is Written on a Dog’s Face?”) personally and via a robust outreach plan, including online and social media platforms and fliers advertising the custom-made project website. Participation from human companions entailed recording and submitting a series of videos of dogs in the home (see below). Human companions gave their written informed consent prior to voluntary participation in the study and were provided updates and opportunities to further engage with the project over the course of the data collection and analyses.

*Participants*. One hundred and eight (108) pet dogs living in households with human companions in North America and Europe were tested. A total of 5 of the 108 dogs submitted were excluded from analyses due to one or more uncodable videos, giving a final total *N* of 103 (Table 1). The canine subjects included various “purebred” and mix-breed dogs, as reported by their human companions. Per the American Kennel Club (AKC) breed group designation [42], there were 11 Working, 7 Toy, 7 Terrier, 24 Sporting, 9 Non-sporting, 16 Herding, 6 Hound, and 23 Mutt/Mixed-breed dogs.

The dogs had to be at least six months of age at the testing time for inclusion in an effort to reduce potential confounds related to the dogs’ early social developmental window (3–4 months of age). The chronological ages of each participant were collected and binned into developmental/cognitive age grades based on the six-category system proposed by Harvey [43]. There were 20 dogs classified as “young” (6 months–2 years), 49 dogs classified as “adult” (2.1–6.9 years), and 34 dogs classified as “senior” (7 years and above). The mean age was 5.2 years (SD = 3.22). The dogs included both males (50) and females (53) and those who were both de-sexed (88) and reproductively intact (15). Forty (40) dogs had no formal training/work status; 31 dogs had basic obedience-level skills; and 32 dogs were highly skilled and/or working dogs, as reported by their human companions.

### 2.3. Experimental Procedure

The data collection period for this study occurred during the COVID-19 pandemic, and thus in-person experimentation was generally not feasible as per the university’s public health protocols. Data collection for a small number (*N* = 20) of subjects was conducted in person, outdoors, though not all of these individuals were included in the final sample (see above). To maximize the total number of study participants and engage the public in community science efforts, the majority of human companions, recruited through interpersonal networking and social media, were given the opportunity to participate remotely via video upload of the dog(s) living in their homes. At-home participants were provided with a study protocol and instructions for uploading their images and videos to secure remote (Dropbox) storage. A potential benefit of in-home data collection is that the test may have more ecological validity by virtue of taking place in the dogs’ natural, day-to-day social and physical environments. 

After filling out a brief demographics survey and behavioral assessment for each canine subject, the dogs’ human companions were instructed to take a photo of their dog(s)’ face(s) and to record four 30-second-long videos of the dog(s) in the following conditions, in the specified order: 

Condition 1: Asocial/Dog at rest—Dog at rest without eye contact from human. 

Condition 2: Eye contact only—Human making eye contact with the dog without speaking, gesturing, or otherwise encouraging a social response. 

Condition 3: Eye contact + Unfamiliar words—Human looking at the dog and speaking in a neutral tone, repeating an unfamiliar phrase twice, slowly.

Condition 4: Eye contact + Familiar words—Human looking at the dog and speaking in a normal to slightly excited tone, using words and/or phrases familiar to the dog, attempting to encourage a social response.

The conditions were designed to elicit the maximum Objective Behavioral Sum (OBS) score (see below) from each individual whose responses to human communication would likely depend on previous experience. The participants recorded their dog(s) in each condition only once unless the dog’s face moved completely out of the frame for more than a third of the video.

For conditions 3–4, humans spoke to the dogs in the language the dog was most used to hearing. The unfamiliar phrase used by all participants was, “*Ancient Egyptians built enormous pyramids to honor the pharaohs. Ruins from many of these sites have been excavated over the years, unearthing mummies, art and relics*”. 

The participants were instructed to keep the front of the dog’s face clearly visible for the duration of each session and to complete filming of all four conditions within 72 h, when possible, allowing at least 30 min between recording different conditions. The participants were asked to locate a quiet, well-lit area of the home to conduct the recording sessions and, where possible, to avoid distractions, such as other humans, dogs, animals, etc. All videos included in the final analysis observed these general instructions. Unfortunately, because some participants uploaded their videos in bulk (i.e., after completing all four conditions), we cannot verify (e.g., using time stamps) the length of time between the condition recordings.

No experimental training phases were required for this study. 

### 2.4. Analyses

Statistical analyses were performed using JASP [Version 0.17.1] and Jupyter Notebook for Python [3.7.15].

The study used a mixed within-between design consisting of 4 conditions (1–4) repeated within-subjects and breed group, age, and sex as the between-subjects variables.

### 2.5. Dependent Measures

*Physical Score (PS)*: The objective measure of physical markings. To assess the complexity of facial physical phenotype and assign a corresponding score, each dog’s face was evaluated using an original matrix that accounted for both pigmentation and perceptible marks/patterning. Perceptible facial marks/patterns included but were not limited to patches (eye or otherwise), “eyebrows”, masks, spots, ticking, “widow’s peak”, and chin strips (Figure 1). The dogs’ facial phenotypes were scored by humans unfamiliar with the individual dogs (i.e., not the human participants).

Per the complexity matrix, a minimum physical score of one (1) would indicate a solid-coated or hyper- “plain-faced” dog; a maximum physical score of nine (9) would indicate a dog with more than two coat colors visible on the face, and at least two markings in each of three facial regions: head/ears, eye area, and mid-lower face.

*Objective Behavior (OB)*: Objective behavioral measures of facial movements (expressivity). The dogs were assessed by independent coders using the Dog Facial Action Coding System DogFACS [44] (see below) in each condition. This measure ranged from 4 (the lowest in any condition across the sample) to 71 (the highest in any condition across the sample).

*Objective Behavioral Sum (OBS)*: The score for each dog was calculated as the sum of the behavioral expressivity scores (OB), as coded according to DogFACS (see description below) across all four conditions (1–4), including the movements for all facial regions indicated (see below). This measure ranged from 41 (the lowest across the sample) to 258 (the highest across the sample). Because the physical score (PS) did not change across conditions, this collapsed measure was used in comparison to the PS. 

*Behavioral Bin (Bin):* The percentile rank of the OBS score for each dog (1–10). This was used in order to compare the OBS to the owner’s subjective expressivity score (see below). 

*Expressivity*: The owner’s subjective evaluation of their own dog’s expressiveness. The owners were asked to rate on a scale of 1–10 their dog(s)’ level of non-vocalizing expression, with 1 = does not seem expressive at all and 10 = very expressive.

*Agreement*: Composite measures consisting of the difference between the owner’s subjective rank of their dog’s expressivity (Expressivity) and the binned objective OBS measure of expressivity, where a score of “0” means the objective and subjective measures are in complete agreement, a negative score indicates the owner ranked expressivity lower than the objective binned OBS, and a positive score indicates the owner ranked expressivity higher than the objective binned OBS. 

### 2.6. Independent Variables

In addition to the dog’s breed group, sex, and age, we performed exploratory analyses on how the PS, OBS, and expressivity measures varied by the following: 

*Eyebrows:* The presence or absence of a physically colored “eyebrow” marking on the face. 

*Time-in-Home*: The duration of time in years the dog had lived in the home with the owner.

*Origin*: The last known place of origin of the dog, as reported by the owner (shelter, rescue, breeder, re-homed, or self-bred).

*Work Status:* Owners reported the level of training/work experience their dog had achieved at test time. Those who had never taken a formal training class were categorized as “unskilled”; those who had completed at least a basic obedience class were categorized as “obedience” level; and those who had one or more training certificates/titles/working dog statuses were considered “skilled”. This included skills in the following areas: agility, rally, conformation, scent work/detection, herding, fieldwork, search and rescue, and service.

### 2.7. DogFACS

The Facial Action Coding System (FACS) [17] is the most widely used and well-regarded tool for measuring facial expressions in human research. FACS is an anatomically based system that describes observable movements of the face in the context of the underlying muscles responsible for the movements. Numbered action units, or Aus, correspond with the visible movements. The system is an effective approach for minimizing experimenter biases related to human emotions and expressions. FACS has been adapted for several animal species (www.animalfacs.com, accessed on 1 November 2020), including dogs. The Dog Facial Action Coding System, or DogFACS [44], is similarly reliable and useful in reducing biases, especially those potentially introduced through anthropomorphizing. Using DogFACS in research requires practice, testing, and certification. Two certified DogFACS coders manually coded the video samples independently, according to the DogFACS manual [44].

The facial areas coded included: 

Upper Face action units (Inner Brow Raiser (AU10), Eye Closure (AU143), Blink (AU145)); 

Mid and Lower Face action units (Nose Wrinkler (AU109), Upper Lip Raiser (AU110), Lip Corner Puller (AU12), Lower Lip Depressor (AU116), Lip Pucker (AU118), Lips Part (AU25), Jaw Drop (AU26), Mouth Stretch (AU27)); 

Mouth action descriptors (Tongue Show (AD19), Blow (AD33), Suck (AD35), Lip Wipe (AD37), Nose Lick (AD137)); 

Ear action descriptors (Ears Forward (AD101), Ear Adductor (AD102), Ear Flattener (AD103), Ear Rotator (AD104), Ears Downward (AD105)); 

Head/Eye action descriptors (Head Turn L/R (AD51/52), Head Up/Down (AD53/54), Head Tilt L/R (AD55/56), Eyes Turn L/R (AD61/62), Eyes Up/Down (AD63/64)). Lip Wipes (AD37) and Nose Licks (AD137) were coded in addition to Tongue Show (AD19), not in place of, where applicable. 

All the above action units and descriptors were included in the calculation of the objective behavioral sum (OBS) score, individual condition scores (OB), and facial region subscores. The miscellaneous behaviors, including sniffing, vocalizing, panting, chewing, licking, and body shakes, were noted but not included in the analyses. An OBS of zero would indicate there were no discernable facial movements with the corresponding DogFACS units in any of the four conditions. The highest behavioral score for anyone canine participant recorded was 258; the lowest OBS for any single canine participant was 41.

### 2.8. Score Validation

All images and video recordings from each canine subject were coded independently by two different DogFACS-certified coders for reliability. There was, generally, concordance between the two scorers; however, if there was an intercoder difference of greater than 5 points (behavioral) or 2 points (physical), those videos/images were rescored. No subjects had to be thrown out due to scorer discordance.



## 3. Results

The preliminary analyses evaluating how sex, the presence of eyebrows, and origin might have impacted the OBS, PS, and expressivity scores found no significant effect. Consequently, these variables were excluded from additional analyses. 

### 3.1. Behavior and Physical Score

To evaluate the effects of various demographic variables on the PS and OBS, a Pearson’s r correlation was used. It included the objective behavioral sum (OBS) score, physical score (PS), age (un-binned), expressivity, time-in-home, agreement, and work status. Several relationships were significant, including OBS and PS (r = −0.326, *p* ≤ 0.001); OBS and age (r = −0.283, *p* = 0.004); OBS and work status (r = 0.289, *p* = 0.003); OBS and time-in-home (r = −0.313, *p* = 0.001); OBS and agreement (r = −0.726, *p* ≤ 0.001); age and work status (r = −0.268, *p* = 0.006); age and agreement (r = 0.268, *p* = 0.006); age and time-in-home (r = 0.876, *p* ≤ 0.001); time-in-home and agreement (r = 0.276, *p* = 0.005). There was a negative trend toward significance between expressivity and PS (r = −0.167, *p* = 0.092). 

The higher the PS score, the lower the OBS score (r = −0.326, *p* ≤ 0.001) (Figure 2); the higher the OBS score, the lower the age (r = −0.283, *p* = 0.004) and the shorter time the dog had lived in their home (r = −0.313, *p* = 0.001). The lower the age, the shorter amount of time in the home (r = 0.876, *p* ≤ 0.001); and the higher the OBS, the less disagreement between expressivity and the OBS (r = −0.726, *p* ≤ 0.001). The more skilled/more training dogs had, the younger they tended to be (r = −0.268, *p* = 0.006) and the higher their OBS was (r = 0.289, *p* = 0.003).

### 3.2. Differences in Behavior across Age and Training/Skill Level Groups

An ANOVA further evaluating the differences in the OBS and age groups and the OBS and work status levels was significant for the age bin [F(2, 94) = 5.5, *p* = 0.005, η^2^ = 0.10], and the post hoc comparisons revealed significant differences between adult and senior dogs, (p_bonf_ = 0.004). Despite the significant correlation noted above, the main effect for work status at three levels (no skill, obedience, skill) was not significant. To increase statistical power, we ran another ANOVA with a two-level version of this factor (no skill–skill). There was a marginally significant main effect with a medium effect size [F(1, 66) = 4.02, *p* = 0.049, η^2^ = 0.050; p_bonf_ = 0.008] (Figure 3). The age-by-skill interaction was not significant [F(2, 66) = 0.227, *p* = 0.797, η^2^ = 0.006].

### 3.3. Behavior between Conditions 

To evaluate the changes in the DogFACS (OB) scores across conditions (1–4), a repeated-measures ANOVA that included a condition (four levels: 1–4) as a repeated measure and breed group as a between-subjects factor produced a main effect for the condition [F(2.8, 285) = 9.06, *p* ≤ 0.001, η^2^ = 0.082], and a between-subjects effect for breed group [F (7, 95) = 2.28, *p* = 0.034, η^2^ = 0.144]. Post hoc pairwise comparisons using the Bonferroni correction indicated that the OB for Condition 1 was significantly lower than for Condition 3 (p_bonf_ = 0.008) and Condition 4 (p_bonf_ ≤ 0.001). The OB score for Condition 2 was significantly lower than Condition 4 (p_bonf_ = 0.001). No other contrast was statistically significant after correction. 

Post hoc comparisons indicated that the breed group population numbers in the sample were insufficient to determine a significant relationship between breed groups.

### 3.4. Movements across Facial Regions between Conditions 

To evaluate the differences in movements within multiple regions of the face across conditions (1–4), a two-factor repeated-measures ANOVA that included face parts (three levels: Ears, Upper Face, Mid Face) and condition (four levels: 1–4) produced the main effects for the face parts [F(2, 204) = 84.99, *p* ≤ 0.001, η^2^ = 0.231], condition [F(3, 306) = 8.65, *p* ≤ 0.001, η^2^= 0.013], and the face parts X condition interaction [F(6, 612) = 11.24, *p* ≤ 0.001, η^2^ = 0.032]. The number of movements in the Upper Face decreased from Conditions 2–3 to Condition 4, whereas the number of movements in the Ears and Mid Face increased from Conditions 2–3 to Condition 4 (Figure 4).

Conditions 3 and 4 (eye contact with humans speaking unfamiliar words; eye contact with humans speaking familiar words) accounted for a higher percentage of the OBS score than Conditions 1 or 2 (baseline; eye contact/no words). 

Interestingly, the majority of action units and action descriptors coded in the Head/Eye and Upper Face regions combined decreased in the percentage of movements per region compared to the OB from Condition 2 to Condition 4 (Condition 2 = 81%; Condition 3 = 76%; Condition 4 = 64%). That is, as the percentage of movements increased across the conditions, so too did the spread of movements across the facial regions in which they were being made (Figure 5). 

### 3.5. Objective vs. Subjective Measures of Expressivity

Human companions were asked to rank their dog(s)’ level of non-vocal expressivity on a scale of 1–10 (1 = “does not seem expressive at all”; 10 = “very expressive”). Nearly half of the people were within a 2-point score deviation (48.5%), and more than two-thirds were within a 4-point score deviation (68.9%) from the dogs’ objective behavior scores. Those who were less accurate (31%) deviated at 5 points difference or more. Notably, the presence/absence of “eyebrow”-like physical markings did not influence human ranking.

Among the most accurate owners (the upper third who were in complete objective–subjective score agreement or with 1 point deviating), eight dogs, or 30% of the dogs, were plain-faced (a PS score of 1), whereas, among the least accurate owners (the lower third, who were 5 points or more deviating), only one dog, or 3% of the dogs had a solid face (Figure 6). 

However, as noted above, there was no significant correlation between the human owners’ subjective measures of their dogs’ expressivity (i.e., expressivity score) and PS and OBS scores. But, there was a trend toward significance between expressivity and the PS, with the higher the subjective expressivity score, the lower the PS (r = −0.167, *p* = 0.092), which we hypothesized may be indicative of an indirect effect of the PS on agreement (the relationship between the objective OBS and subjective expressivity)—the OBS was higher for dogs with lower PSs, while objective and subjective scores of dogs with higher OBSs were more closely aligned. For adult dogs only, owners’ subjective assessments of their dogs’ expressivity were more aligned with their dogs’ objective behavioral expressivity scores when the dogs had a lower PS or fewer physical markings on the face. That is, owners of adult dogs gauged their dogs’ expressivity more accurately if the dog had a plain face than if the dog’s face was more complex (Figure 7).

## 4. Discussion

It is somewhat surprising that this study finds that the markings and coloration on dogs’ faces have a similar effect on the perception of their facial expressions as do the markings on the faces of social primates (that is, plainer faces are seen as more expressive). Dogs display a striking convergent evolution with non-human primates in regard to the diversity of facial hair patterns and ornamentation, including and especially such markings as “eyebrows” and “widow’s peaks” [45], color(s), and furnishings. However, while these phenotypes are naturally occurring in non-human primates, they are artificially selected in dogs. The history and nature of intentional breeding for dogs who are adept at performing distinct tasks within human society would, by default, also necessitate that dogs of all different physical phenotype variations have the ability to communicate well with humans (including and especially attending to human faces). 

Therefore, while statistically significantly correlated in this study, facial markings and coloration probably do not have a real biological effect on dogs’ capacities for facial movements; rather, the significance may be the result of movement being the salient signal to human observers. Indeed, dogs seem to have adapted their behavioral features of the face significantly to communication with humans, regardless of the influence of physical features, and have also developed early emerging social skills to prepare and allow for cooperative communication with humans [46,47,48,49].

Because the physical features of faces may not be as important to conspecific communication in domestic dogs as they are for group-living/hunting wolves and non-human primates [30,34]—see Gergely et al. [50] and Mongillo et al. [51]—it is reasonable to assume that dogs of any breed/mix would behaviorally overcome any natural and/or artificial selection by humans for specific physical traits. All breeds possessing the same potential primacy of productive rather than passive (e.g., markings) communication toward humans can be considered a novel adaptation unique to the demands of being dependent in an interspecies relationship. Our findings that dogs who were skilled and certified working dogs were more behaviorally expressive in their facial movements than those with no skill/training (Figure 3) support this hypothesis—attention and response to humans during episodes of social learning, cooperation, human–dog coordinated action, and other such communication-heavy activities that working dogs engage in would suppose an increased use of facial gestures.

Likewise, differences in the objective expression scores (OBS) between adult and senior dogs (with senior dogs making significantly fewer expressions) could be the result of senior dogs being generally less physically mobile/physically uncomfortable or may suggest that (a) diminished cognition slows response [52,53], or (b) older dogs have learned that a higher rate of gestural expression may not be necessary to convey their intended signals to a familiar human partner (they do not need to “try” as hard).

Although we lacked the power to determine what the significant breed group differences were, it is unsurprising that they should exist. Given the dramatically different working and companionship roles of the dogs included in the groups, it is reasonable that their facial communication strategies would differ [54,55,56,57]. For example, while sporting breeds have historically been bred to work alongside hunters in the field, pointing and retrieving (mostly without auditory signaling so as not to alert their prey), non-sporting dogs breed more diverse social/working backgrounds, and toy breeds are those that have been selected strictly as companion animals.

Regarding the within-subject variation, we found that as the information from the human companion changed, the response from the dogs changed—different parts of the face seemed to serve different functions in different conditions of human attention (Figure 5). 

As Conditions 3 and 4 were conditions under which human companions were speaking to the dogs, it is reasonable to assume that the increase in expressive behaviors in these conditions was related to the increased attention and responsiveness from the dogs toward the human companions in a more social context.

Discounting the resting state condition (Condition 1), the majority of movements compared to the OB in Conditions 2 and 3 occur in the Head/Eye + Upper Face (including Brow Raiser, Blink, Eye Closure, Eye Movements Up, Down and Left and Right, and Head Movements). Recall that Condition 2 involved a human companion making eye contact but not speaking, and Condition 3 involved a human speaking unfamiliar words. These were conditions under which the canine subjects may have been confused and/or awaiting further instruction/clarification. Infant developmental literature suggests that a similar phenomenon occurs with human infants when faced with similarly ambiguous cues or an attentive but still face from an adult caregiver, wherein the infants typically decrease expressive behavior and even gaze and often become stressed when presented with a still face [58,59]. In a recent pilot study examining the still-face paradigm in dogs, Barrera et al. [60] reported a decrease in affiliative behaviors in dogs toward humans during the still-face phase. 

Condition 4, on the other hand, has a much greater distribution of movements across facial regions. In this condition, in addition to paying attention to the humans, dogs may have been provoked by familiar and exciting words and phrases to respond using a broader gestural repertoire (Figure 4 and Figure 5). 

Finally, humans characterized their own dog(s)’ level of expression with moderate accuracy, though, of particular interest to those hoping to enhance the human–dog bond, human companions in this study tended to overestimate their dogs’ expressivity—only 22 respondents (~20%) scored their dogs as less expressive than the OBS score indicated—indicating perhaps a confound between dogs’ responses during experimental conditions and their “everyday” behavior, or else over-eager interpretation. However, according to Sullivan et al. [61], humans are better at categorizing canine facial displays of emotion than they are at categorizing those of chimpanzees or bonobos. Those authors attribute this skill discrepancy to the fact that, although *Pan* looks more human-like, we are more *socially familiar* with dogs. 

In reviewing the data here, human companions may characterize dogs’ rates of facial expressivity with more accuracy if the dog has a plainer face independent of emotional valence (we found this was the case for owners of adult dogs, see Figure 7). Conversely, Bloom et al. [62] suggest that people are able to identify emotions across all breeds at a rate higher than expected by chance, except for Dobermans (a plain-faced breed), which they hypothesize is due to the breed’s dark facial color, which may obscure expressions.

Among additional future studies, our results encourage further investigation of human perception of dogs’ facial expressivity as humans may be transferring an entrained preference for reading the relatively plain faces of human conspecifics to our interactions with canines. Although human facial morphology and relative muscular innervation support some of the most complex facial expressions [63,64], we are relatively plain-faced compared to many social, group-living primates (e.g., guenons, callitrichids) [34], and individual differences in superficial facial features (though not movements) may be used more for identification than communication [65,66]. Markings and patterning potentially obscure the behavioral features on dogs’ faces for the humans looking at them, and thus a solid-coated face would seem to be more expressive simply because there is less visual “noise”.

Of course, it would also be worth repeating this study without some of the limitations imposed by pandemic-era data collection. Primarily, a larger sample size, including an equal number of participants from each breed and age group (especially given the behavioral differences observed among these groups), would be of value. A larger sample size would also aid in controlling for commonly observed canine facial features, such as ear position, brachycephaly, wrinkling (e.g., one participant was a Shar Pei), long/shaggy hair around the eyes, and variations in muscular robusticity, which may have contributed to skew. Comparing the results to those using data collected in a controlled laboratory setting where processes could be standardized would also be of interest, as technical challenges arising from at-home recordings could skew the visibility of facial expressions. This would be especially pertinent as community science solicitations become more widely used for data collection.

## 5. Conclusions

In our study of analyzing the facial expressivity and physical characteristics of more than 100 companion dogs (*N* = 103), we found that dogs with plainer faces (fewer markings and/or colors) appear to be more behaviorally expressive in objective measures. Among the age groups, adult dogs are more expressive than senior dogs, and dogs that are highly skilled are more expressive than those who have had no training or working experience. Especially relevant to interspecies communication and cooperation, dogs respond with movements more evenly distributed across multiple facial regions when responding to familiar words and tones from humans than from ambiguous or asocial cues; humans tend to be more accurate at judging the expressivity of dogs with plainer faces. 

The domestication of dogs and their coexistence with humans has influenced the biological and social development of both species. While the suite of physical changes that now separates dogs from extant wolves has largely been selected for by humans, studies like this one suggest that some changes may not have been as deliberately cultivated as others. The results from this study suggest that there may even be underlying, conserved preferences for certain facial features that humans have unwittingly selected for similar reasons that we may find one human more or less attractive, trustworthy, “easy to read”, or any number of other traits. 

Understanding how and to what degree biases such as these and other interactions with humans (including the potential projection of human biases onto dogs) impacts the development of novel modes of communication in dogs could provide valuable insight into what shaped early human culture. Likewise, and perhaps more importantly, by gaining a fuller view of how dogs communicate with humans and how we receive and perceive their efforts, we can be better equipped to support them in the critical roles they fill within our society.

Indeed, as the field of canine science expands, findings from studies such as this offer new insight into understanding and navigating the continuously evolving relationship between humans and dogs and will hopefully also prove useful in exploring new avenues of research among a myriad of other taxa and social systems.

## Figures and Tables

**Figure 1 animals-13-02385-f001:**
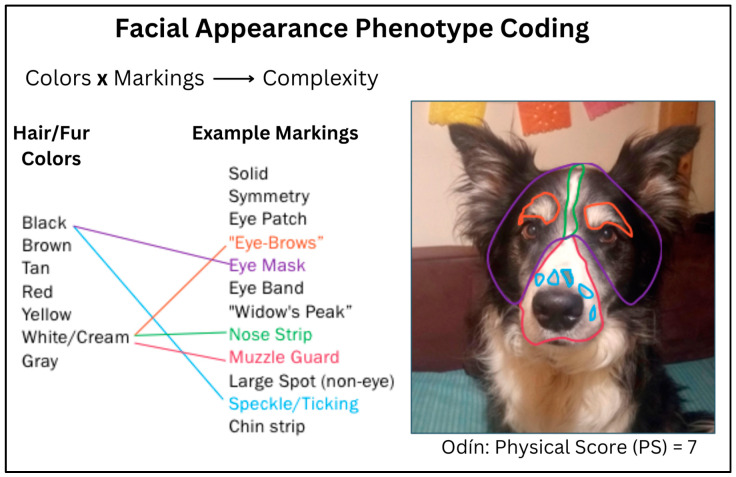
Facial features that contributed to a dog’s overall complexity score (PS) included both color and markings, such as those highlighted here. Odín, a border collie who participated in the study, provides an example of a dog with a physical score (PS) 7.

**Figure 2 animals-13-02385-f002:**
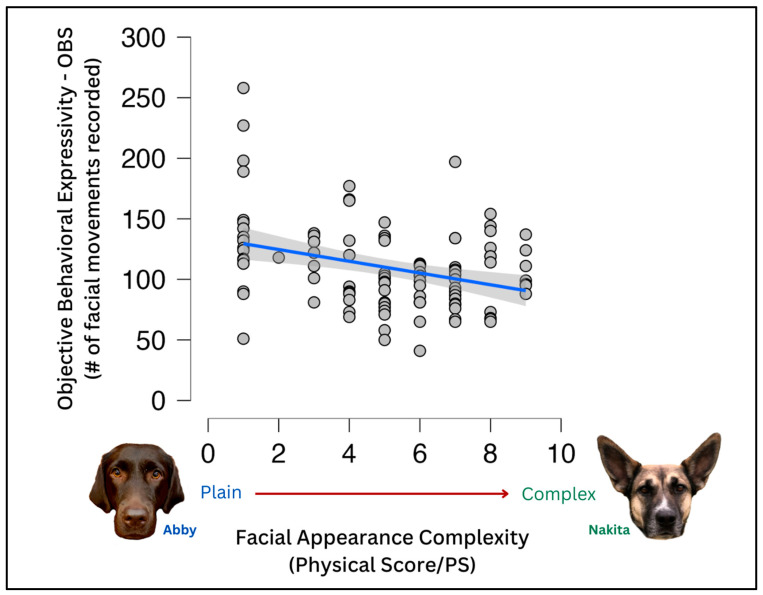
Relationship between OBS and PS. There was a slight though statistically significant correlation between the objective behavioral sum (OBS) score and appearance or physical score (PS). Plainer-faced dogs had slightly higher OBS scores than those with more complex faces.

**Figure 3 animals-13-02385-f003:**
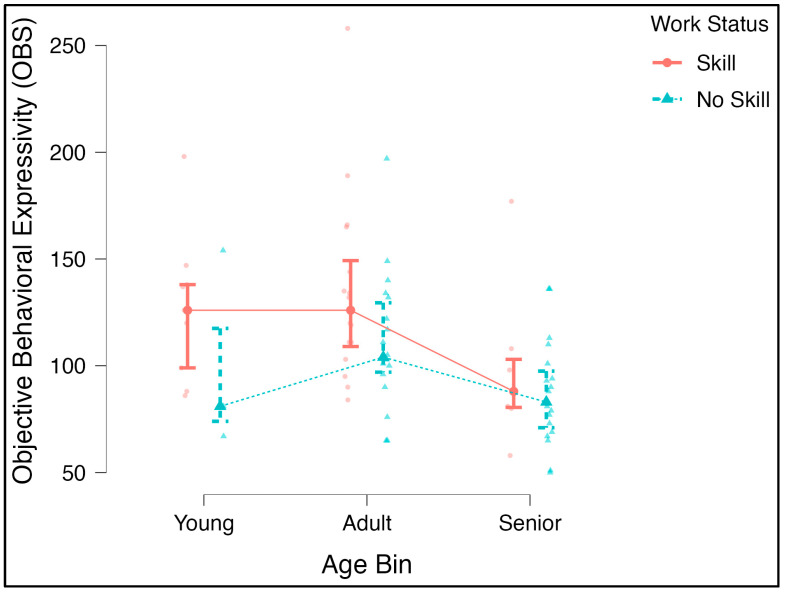
OBS and Age Groups. On average, adult dogs had a significantly higher objective behavioral sum (OBS) score than senior dogs. Dogs who were reported by owners to be “skilled”, that is, had one or more training classifications or advanced work status above basic obedience, received a higher OBS than those dogs who had no training/status.

**Figure 4 animals-13-02385-f004:**
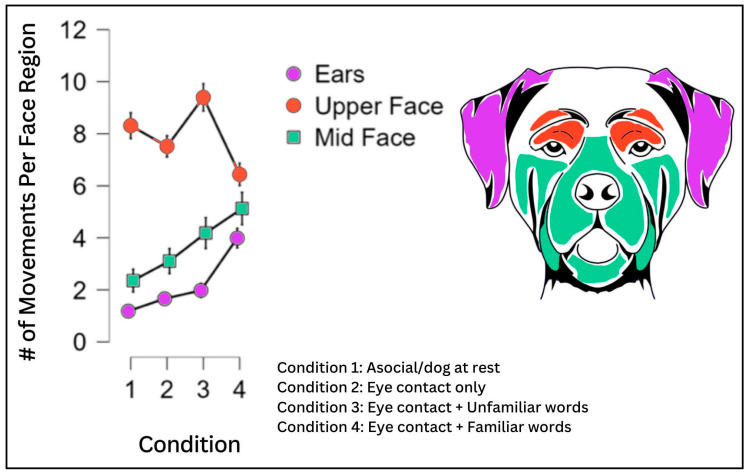
Face Movements Across Conditions. Movements of the Upper Face were higher in Conditions 2 and 3, wherein humans stared at dogs and said nothing (Condition 2), or else used unfamiliar words (Condition 3), than in Condition 4, where humans made eye contact and used familiar words. Condition 4 provoked a more equally distributed spread of movements across facial regions.

**Figure 5 animals-13-02385-f005:**
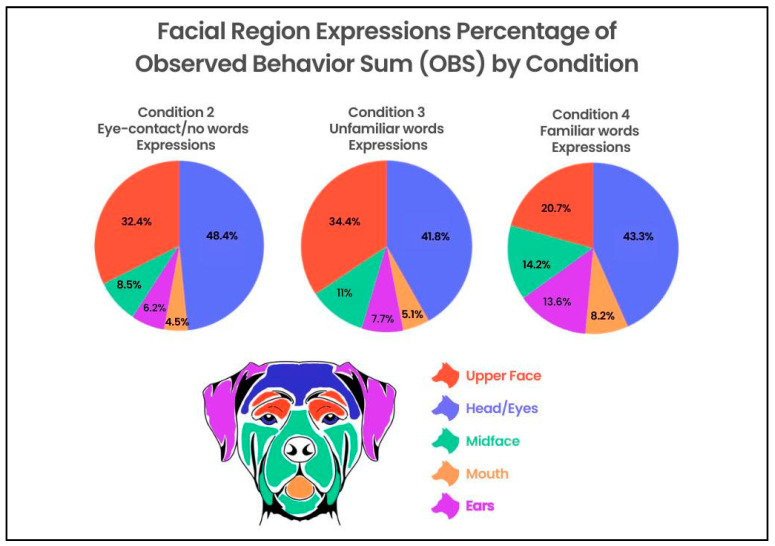
In Conditions 2 and 3, humans stared at dogs and said nothing (Condition 2) or else used unfamiliar words (Condition 3), provoking movements in the head/eye region, which may be related to anticipatory gaze holding. Dog facial movements are more broadly distributed across facial regions when responding to humans who are speaking in familiar words and tones (Condition 4).

**Figure 6 animals-13-02385-f006:**
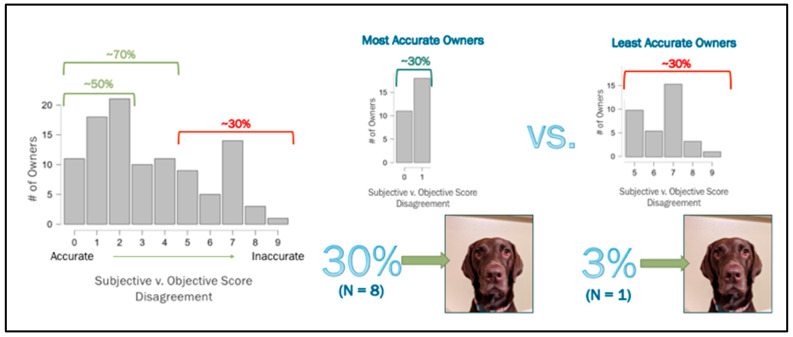
There was a higher percentage of human companions who had a plain-faced dog and were accurate when subjectively judging their dog’s expressivity than those who had a plain-faced dog and were inaccurate.

**Figure 7 animals-13-02385-f007:**
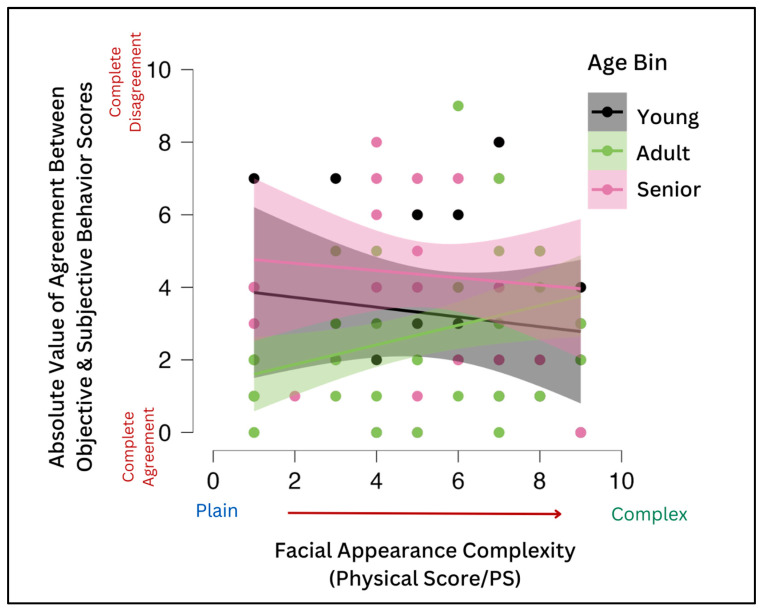
Age as a Determinant of Agreement. While there was no correlation between the physical score and subjective expressivity ranking, owners of adult dogs gauged their dogs’ expressivity more accurately compared to objective scores (OBS) if the dog had a plain face than if the dog’s face was more complex.

**Table 1 animals-13-02385-t001:** Summary of dogs included in study sample.

	Classification	*N* Dogs in Sample
AKC Breed Group	Working	11
	Toy	7
	Terrier	7
	Sporting	24
	Non-sporting	9
	Herding	16
	Hound	6
	Mixed-breed	23
Age Bin	Young (6 months–2 years)	20
	Adult (2.1–6.9 years)	49
	Senior (7+ years)	34
Sex	Male	50
	Female	53
Reproductive Status	De-sexed	88
	Intact	15
Training/Skill Level	Unskilled/No training	40
	Basic obedience	31
	Skilled	32

## Data Availability

Video data may be available upon request in accordance with privacy considerations. Subject coding data are available publicly via Mendeley Data: Sexton, Courtney (2023), “Canine Facial Phenotypes and Expressive Behaviors”, Mendeley Data, V1, doi:10.17632/br92x9768y.1.

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
