# Peer review of "What Is Written on a Dog’s Face? Evaluating the Impact of Facial Phenotypes on Communication between Humans and Canines"

_animals, 2023, doi:10.3390/ani13142385_

Round 1
Reviewer 1 Report
This manuscript presents important data on how humans and dogs communicate via facial movement, focusing on how facial fur pattern may influence human processing of dog facial movement. These are important data as they pertain directly to the co-evolution of humans and dogs. I have some comments for improvement of the manuscript, but they are rather minor.
1) Describe how you settled on the age groups you used in the study. How did you determine that the ages corresponded to "young", "adult", and "senior".
2) The complexity scores do not include ear shape. This is important since we know that humans broadly consider ear shape in general decisions about whether a dog seems "friendly" and ear movement is very important to humans. I suggest the authors go back and include ears in their scoring system.
3) The figures are generally informative and could almost stand alone without the accompanying text. That said, Figure 4 has a lot of scatter associated with it. Could the authors consider that the lower scores with young dogs may reflect the short amount of time that the human and dog have been together? In other words, is there an "experience" effect associated with the lower scores in Figure 4? Do humans need time with their dogs in order to score well? The lower effect on senior dogs is intriguing -- do the authors have any information on those dog subjects' eye health? Did glaucoma or other changes in the dogs' eyes potentially impact the scores?
4) There are so many data in this manuscript that I had to read it 3 times in order to suss it out. These are great data and I don't have a specific suggestion but can the authors somehow create a box that summarizes all the data together?
5) One of the things that we wonder about in dog evolution is how short-faced breeds compare to dogs with average/long faces. Can you make any comments about how humans score those short-faced breeds?
6) This is likely a comment that the authors can address in future studies, but it would be valuable to replicate this study but with dogs interacting with dogs. Can humans accurately score dogs when they are not interacting with humans?
Generally, this is a very well done study and provides valuable data. Well done!
Anne Burrows
Author Response
Dear Dr. Burrows,
Thank you so very much for taking the time to review our manuscript, and for your thoughtful feedback. Your work has been of great inspiration to this study. Please see our responses to your individual comments in the attached Word document.
Sincerely,
Courtney Sexton

Reviewer 2 Report
The evaluation of facial expressions is a topic that has become increasingly relevant in veterinary medicine, particularly for companion animals. The present study shows relevant information regarding this issue. A general comment would be to clearly state the aim of the study and improve the conclusions.
Line 15: Please, revise the Instructions for authors to follow the format of the journal (e.g., adding a simple summary, amending in-text citation style – [1], [1-3], Wood et al. [13]–, and check if the section “highlights” is required in the main text).
Lines 19-20: Clearly state the aim of the present study. For example, “This study aims to evaluate the potential impact of superficial facial markings….”
Line 22: Mention additional information such as the sex and age of the animals.
Lines 23-24: Consider adding the p-value and the correlation value when mentioning statistical significance and strong/weak correlations.
Line 29: Add a general conclusion for the study.
Lines 64-67: After this sentence, I would recommend adding the concept of “facial expression” and its association to communicate and transmit emotions.
Line 75-83: I consider it relevant to mention that dogs can not only respond to human facial expressions but can also develop their facial expressions to several stimuli, and this might be the consequence of domestication and cohabitation that the authors are mentioning. Lines 102-104 could be moved here and these articles might be helpful to support this information: https://doi.org/10.3390/ani11113334 and https://doi.org/10.1371/journal.pone.0082686. Also, it would be interesting to mention a little more detail on the differences between wolves and domestic dogs, together with the paedomorphic facial expression that is distinctive in dogs (included in the articles that I recommended above).
Lines 104-130: I suggest summarizing and re-structuring these lines to maintain order. Lines 104-108 seem like the aim of the study, but then the hypotheses follow, and on lines 125-130 the aim seems to be mentioned again.
A suggestion could be linking and moving the research questions in lines 125-130 after line 101. Then stating the aim of the study and the hypotheses.
Line 163: Regarding the inclusion/exclusion criteria, did the authors exclude animals with ophthalmologic or neurological disorders that could have altered their facial expressions? Or do the authors consider only clinically healthy subjects? Also, was the coat color an inclusion factor?
Line 178-180: Since the recording process was performed mainly by the owners in their homes, additional instructions were given regarding the recording distance, quality of the video, and other technical issues. I comment on this because the quality of the video could greatly influence facial expression recognition by making it difficult to recognize certain facial movements.
Line 213: Please, include the significance level, as well as the statistical model to compare variables and the method to obtain the correlations. This section needs more detail into the test that the authors used for each variable.
Figure 2 I recommend deleting the DogFACS screenshot on the left side of the figure. Leaving only the photographs with the FAU (and maybe adding some more) is enough to understand that the authors used Dog FACS.
Line 419- 424: Although this information is relevant, I recommend starting the discussion by mentioning the main findings of the present study.
Line 439: Some other adapted facial features could be discussed in these lines.
Lines 454-461: I consider it relevant to discuss not only the age of the animals but also about external features such as color of the face, natural position of the ears –floppy or upwards ears–, and other breed-related factors (e.g., wrinkly skin from some breeds) that might affect how we humans perceive their facial expressions (https://doi.org/10.1016/j.beproc.2022.104752, among other studies).
Line 502: Briefly mention some limitations of the present study. Additionally, a future application of this study could be into understanding facial expression in dogs to determine pain or health issues (https://doi.org/10.14573/altex.1607161, https://doi.org/10.1079/PAVSNNR201914028, and https://doi.org/10.48550/arXiv.2206.05619). The authors could include this.
Line 503: Consider summarizing the conclusion. As it is now, it seems a little bit too long. The authors must be able to resume and highlight the main findings.
Author Response
Please see the attachment, thank you!

Round 2
Reviewer 2 Report
The authors have made changes to each comment. The article now has important improvements.
I have no additional comments to include. I suggest it be published.